# Cost of Illness Analysis of Type 2 Diabetes Mellitus: The Findings from a Lower-Middle Income Country

**DOI:** 10.3390/ijerph191912611

**Published:** 2022-10-02

**Authors:** Muhammad Daoud Butt, Siew Chin Ong, Muhammad Umar Wahab, Muhammad Fawad Rasool, Fahad Saleem, Adnan Hashmi, Ahsan Sajjad, Furqan Aslam Chaudhry, Zaheer-Ud-Din Babar

**Affiliations:** 1School of Pharmaceutical Sciences, Universiti Sains Malaysia, Penang 11800, Malaysia; 2Consultant Diabetologist, Umar Diabetes and Foot Care Centre, Umar Diabetes Foundation, Office 1, Executive Complex, G8 Markaz, Islamabad 46000, Pakistan; 3Department of Pharmacy Practice, Faculty of Pharmacy, Bahauddin Zakariya University, Multan 60800, Pakistan; 4Department of Pharmacy Practice, Faculty of Pharmacy & Health Sciences, University of Baluchistan, Quetta 87300, Pakistan; 5Ibn Sina Community Clinic South Wilcrest Drive, Houston, TX 77099, USA; 6Department of Pharmacy, Quaid-i-Azam University, Islamabad 45320, Pakistan; 7Department of Pharmacy, University of Huddersfield, Huddersfield HD1 3DH, UK

**Keywords:** cost of illness, diabetes mellitus, direct cost, indirect cost, diabetes in Pakistan, diabetes economic burden

## Abstract

Background: Diabetes is a major chronic illness that negatively influences individuals and society. Therefore, this research aimed to analyze and evaluate the cost associated with diabetes management, specific to the Pakistani Type 2 diabetes population. Research scheme and methods: A survey randomly collected information and data from diabetes patients throughout Pakistan out-patient clinics. Direct and indirect costs were evaluated, and data were analyzed with descriptive and inferential statistics. Results: An overall of 1839 diabetes patients participated in the study. The results have shown that direct and indirect costs are positively associated with the participants’ socio-demographic characteristics, except for household income and educational status. The annual total cost of diabetes care was USD 740.1, amongst which the share of the direct cost was USD 646.7, and the indirect cost was USD 93.65. Most direct costs comprised medicine (USD 274.5) and hospitalization (USD 319.7). In contrast, the productivity loss of the patients had the highest contribution to the indirect cost (USD 81.36). Conclusion: This study showed that direct costs significantly contributed to diabetes’s overall cost in Pakistan and overall diabetes management estimated to be 1.67% (USD 24.42 billion) of the country’s total gross domestic product. The expense of medications and hospitalization mostly drove the direct cost. Additionally, patients’ loss of productivity contributed significantly to the indirect cost. It is high time for healthcare policymakers to address this huge healthcare burden. It is time to develop a thorough diabetes management plan to be implemented nationwide.

## 1. Introduction

Diabetes is among the most common non-communicable diseases in emerging and developed nations. Substantial literature indicates the high prevalence of diabetes in low- and middle-income countries [1]. As per the International Diabetic Foundation (IDF), it has also been found that approximately 537 million people have diabetes, and it is expected to increase by 643 million in 2030 [2]. In addition, study indicated that 75% of people between the ages of 39 to 70 years and living in middle and low-income countries have higher risk of diabetes. In Pakistan, it was found that 6.9% of adults have diabetes, and it is expected to increase by 8.2% in 2040 [3,4,5]. However, there are increased healthcare expenditures and higher costs associated with diabetes management. It has been reported that a significant portion of the diabetes cost is associated with complications and macro-vascular disease that frequently occur due to type 2 diabetes [6,7]. According to the Asia Development Bank (ADB), the poverty rate in south Asian countries was 15.09%, and the total extremely poor population was 256.24 million people. Furthermore, the poverty rate in Pakistan was 7.93%, with 10.26 million people living in extreme poverty. [8]. Additionally, in Pakistan there are deficiencies in healthcare system, and there is no universal health coverage (UHC) for the population. Therefore, the increasing cost of diabetes has become a global concern in many nations, especially in low-income countries [9]. However, this rise in the number of diabetes has increased healthcare costs, increasing productivity loss and burden on the economy. Some previous studies have emphasized that the high cost of type 2 diabetes in healthcare organizations has increased the overall loss of economic activities and productivity [9,10,11,12,13].

The management of diabetes has gained international attention, and several countries—including Saudi Arabia (21%), Sri Lanka (16%), Malaysia (16%), Mexico (15%), and the US (16%)—spend a significant amount of their health expenditures on diabetes care [14]. The American Diabetes Association’s cost estimates for the US population are notable instances of cost-of-illness research; the total burden for 2012 was projected to be USD 245 billion, taking into account higher health spending and productivity losses caused by diabetes. The relevance of labor market consequences for the overall economic burden was highlighted by the fact that indirect expenses accounted for 28% of the total costs of diabetes in this research [15].

There has only been a few cost of illness studies being conducted in Pakistan. The study by Khowaja et al. reported the direct cost of diabetes management at USD 197 in 2007. It was also estimated that medication and laboratory investigations contributed to the highest direct cost incurred at 46% and 32%, respectively [13]. Another study reported in 2018 that the total direct cost of diabetes management is USD 332, with 60.4% accounting for the total medication cost of management. In these studies, there was a lack of indirect cost computation; therefore, no indirect cost data are available for the region [4]. A study in Bangladesh reported that the overall cost of diabetes management was USD 864.7, with 60.7% accounting for the direct medication cost [16].

To be sure, developed nations are not the only ones where there is evidence of substantial economic consequences. For instance, Seuring and colleagues examined the findings of 86 cost-of-illness studies published between 2001 and 2014 and discovered evidence of a significant economic burden in low-income and middle-income countries (LMICs), with annual direct costs ranging from USD 242 to 4129; (2011 purchasing power parity) per capita and indirect costs ranging from USD 45 to 16914 per capita. It is challenging to compare cost-of-illness studies within and between nations, since the costs included in the calculations and the methodologies used to quantify costs differ significantly in the literature in circulation [17].

Studies on the “cost of illness” (COI) explore the effects of a disease on people, communities, and the nation from various angles. COI research aims to identify and estimate all direct, indirect, and intangible costs associated with a specific condition. The output estimates the societal financial costs associated with a particular disease. It is widely acknowledged that determining the total financial burden of a disease is a valuable tool for developing national and international health policies [18].

Approximately 12% of healthcare expenditures are attributable to diabetes management, which is about USD 827 billion globally [19]. Most studies have highlighted a substantial increase in the number of people living with diabetes, along with the rise in poverty [20,21,22,23,24]. Moreover, this is coupled with the fact that medicine prices have increased substantially in Pakistan over the past two years [25,26]. Therefore, the current study aimed to assess the cost of illness of diabetes. Pakistan is a lower-middle-income country with limited healthcare resources, and this study will help the healthcare regulatory bodies and policymakers to optimally allocate resources to manage diabetes.

## 2. Materials and Methods

### 2.1. Study Design and Setting

The current multi-center study assessed the direct and indirect healthcare costs of patients with type 2 diabetes (T2D) in Pakistan. The study was conducted between October 2021–February 2022. The study setting was outpatient clinics dealing with diabetes.

### 2.2. Study Population and Sampling

The study population was patients diagnosed with T2D. T2D patients were included if their age was ≥ 18 years, did not have any malignancies, and did not have other chronic diseases or drug abuse. The prevalence-based sample size was calculated for the current study. The sample size was calculated using OPENepi online sample size calculation software. A sample size of 1782 ≈ 1800 was calculated based on the 12% prevalence of T2D [26], with 20% relative precision, an odd ratio of 1.8, and 80% power [27].
$$N_{\mathrm{Kelsey}}=\;\frac{{\left(z_{\alpha /2}\;+z_{\beta }\right)}^{2}\;p\left(1-p\right)\left(r+1\right)}{r{\left(p_{0}-p_{1}\right)}^{2}}$$

Pakistan is a lower-middle-income country comprised of four provinces, Punjab, Sindh, Baluchistan, and Khyber Pakhtunkhwa, and the sample size was distributed equally (each = 450). Three districts were selected randomly from each province, and the respondents were recruited equally from each section (each district = 150 participants). The study participants were recruited through cluster sampling, and three clusters from each district were collected (50 participants per cluster). All metropolitan cities in Pakistan were randomly selected (including rural and urban settings) within each province were included in the study sample, which was chosen using a stratified two-stage cluster design. The patients were recruited by seven teams of trained field workers overseen by three project leaders. The field team gathered basic demographic information and asked the selected sample’s consent to provide pertinent facts and answer all the questions about the resources utilized.

### 2.3. Study Tool and Data Collection

A self-developed structured questionnaire was used for data collection (Appendix A). The developed questionnaire was incorporated into an online server using google documents, the secured web-based application used for data collection. Before the main study, a pilot study was conducted with 35 participants to assess the internal consistency of the questionnaire. A Cronbach alpha value of 0.87 was obtained, indicating a good internal consistency. After obtaining informed consent from patients selected randomly in the diabetes outpatient clinics, patients were invited to complete the structured questionnaire online (Figure 1).

The questionnaire comprised of six different sections. Section 1 comprised personal identifiers (e.g., which hospital the data were collected from). Section 2 comprised sociodemographic and general information. Section 3 contained health status and behavior information and Section 4 with co-morbidities/complications and treatment information. Section 5 and Section 6 comprised treatment regime information and diabetes cost information, respectively.

The patient’s medical records from the clinics were also reviewed to retrieve the patients’ clinical records and the number of services received throughout the preceding year from the recruitment date. Moreover, the current prescribed medicine data were obtained for the prescription. The data were collected using an electronic database, and further information was added to the final database.

### 2.4. Ethical Approval and Consent to Participate

The Bio-Ethics Committee (BEC) of Bahauddin Zakariya University, Multan, permitted this prospective observational study in Tertiary Care Hospitals & Diabetes Outpatient Clinics (ACAD/EXT/01/2022).

Written informed consent was taken from all the participants included in the study without assistance from others; in the case of illiterate individuals, consent was obtained from their legal guardians. The study was conducted as per the Declaration of Helsinki. The current study was reported according to “The Strengthening the Reporting of Observational Studies in Epidemiology (STROBE)” guidelines.

### 2.5. Calculation of Cost

#### 2.5.1. Direct Medical and Non-Medical Cost

The direct cost was premeditated using a bottom-up approach. It was divided into two categories: direct medical costs, which include the costs of hospitalization, out-patient visits, medicine, laboratory tests, and other service utilization (including the use of self-monitoring blood glucose and consumables); and direct non-medical costs, which include the costs of transportation, hotel stays (as hospitals are located far from home), and meals (special diet plan, as recommended by international diabetes federation).

To compute the direct cost, the micro-costing approach was utilized to identify cost items in as much detail as possible. Each medicine and therapy’s cost per day was calculated. The costs of treatment, consulting, and laboratory tests that patients paid out of pocket (OPP) in the selected regions were obtained from a tertiary-level hospital. Private health insurance is limited in Pakistan. Hence, patients were responsible for all costs. Individually, participants’ replies to the questionnaire were used as a reference for other components of direct medical and non-medical costs. The unit cost was multiplied by the number of medical services supplied the previous year to determine each direct cost component. The hospital bill for each patient was calculated using their hospital record, which the hospital’s accounts department provided. All direct medical and non-medical expenditures were added to the total direct cost.

#### 2.5.2. Indirect Cost

The indirect cost for patients and their caregivers was calculated on the way to the hospital. The productivity time lost between out-patient visits and during hospital admission was documented based on information provided by patients and their caretakers. The indirect cost for productive workers in the formal workforce or housewives was calculated using the human capital approach but not for people who were unable to work (retired or ill-health) or who chose not to work. The productivity loss of housewives was calculated using the minimum wage rate of PKR 120 (USD 0.7) (USD 2484/annum) set by the court, as well as the median income of the participants in the formal workforce. Total direct and indirect costs were added together to determine the total cost.

All expenditures were computed in Pakistani rupees (PKR), then converted to USD using the mid-year currency conversion rate for 2022 (USD 1 = PKR 180).

#### 2.5.3. Data Analysis

The data from the online database were imported into Microsoft Excel. The data were thoroughly checked for errors and then exported into SPSS. Data were analyzed using SPSS v25. The categorical data were presented as frequency and percentages, while the continuous data were tabulated in mean (standard deviation, SD) and median. Using inferential statistics, the study variables’ significance was established (Mann–Whitney U test and Kruskal–Wallis tests). Pearson correlation coefficient analyzed the relationship between variables. A median regression analysis was implied to evaluate the contributing variables to the direct and indirect costs. The 95% confidence interval (CI) of regression coefficients was determined by bootstrapping. The *p*-value ≤ 0.05 was considered significant.

## 3. Results

A total of 1839 patients were recruited in the current study. Most patients were 40–50 years (60.1%), and more than half were female (52.2%). Moreover, more than one-quarter of the participants had graduated from college and above (31.1%), and 53.4% were unemployed. In addition, most of the participants were married (88.7%). The detail of the other demographics and clinical characteristics of the study population can be seen in Table 1.

Most patients were on oral antidiabetic drugs (95.3%), followed by insulin (12.0%), and then followed by a combination of oral antidiabetics (Metformin, Sulfonylureas, Dipeptidyl Peptidase 4 (DPP-4), Sodium Glucoside-like Transporters 2 (SGLT 2) and insulin (44.8%)). The patients using higher cost Glucagon-like peptide-1 (GLP-1), a new-generation insulin (NGI) group, were few (3.4%), as shown in Table 1.

### 3.1. Direct and Indirect Cost of Diabetes Management Care

The cost of medication (USD 274.5 ± 8.5) and consultancy (USD 16.51 ± 3.23) were the main contributors to the direct cost of diabetes management. Moreover, the travel (USD 4.3 ± 2.98) and investigation costs (USD 12.32 ± 1.58) also had a high share of the direct cost of diabetes management care. In terms of indirect cost, the loss of production was observed higher in the office/businessmen population (USD 106.58 ± 2.5), followed by laborers (USD 15.08 ± 1.4). The detail of direct and indirect costs can be seen in Table 2. To compute the cost of illness for diabetes population the estimated diabetes prevalence of 26.7%. The estimated burden of diabetes cost of management turns out to be USD 24.39 billion.

### 3.2. Socio-Demographic Correlation with Cost of Care

Regarding the correlation of socio-demographic characteristics with cost of diabetes care, gender (direct cost: r = 0.78, *p* = 0.03, indirect cost: r = 0.52, *p* = 0.03), and duration of diabetes (direct cost: r = 0.098, *p* = 0.003, indirect cost: r = 0.78, *p* = 0.005) had a positive correlation with direct and indirect cost. However, the monthly household income negatively correlated with the direct cost of diabetes care (r = -0.92, *p* = 0.002), as shown in Table 3.

The mean direct cost was observed to be high in patients, having an age of greater than 60 years (USD 1376.9), secondary education (USD 874.1), retired working status (USD 1027), and patients having more than three co-morbidities (USD 1234.3). Moreover, patients with a disease duration ≥ 11 years (USD 1044.1), poor glycemic control (567.9), dyslipidemia, and hypertension history (USD 926.9) also had a higher mean direct cost. Whereas the patients with a duration of diabetes ≥ 11 years (USD 116.7), good glycemic control (USD 84.5), history of hypertension (USD 94.3), and three or more complications (USD 117.1), had a high share of indirect cost. The detail can be seen in Table 4.

The total mean cost of diabetes care was USD 740.1, amongst which USD 646.4 for the direct cost was cost-shared chiefly by the medicine (USD 274.5) and hospitalization (USD 319.7), and USD93.7 was the indirect cost, which was primarily shared by the productivity loss by the patients (USD 81.3) (Table 5). The total cost without hospital admission (n = 1365) was USD 319.6, among which the direct cost shared USD 275.1, and the indirect share was USD 44.5.

The median regression analysis showed that patients with an age greater than 60 years (*p* < 0.001), on both oral antidiabetic drugs and insulin (*p* < 0.001), with poor glycemic control (*p* = 0.009), and with multiple co-morbidities (*p* = 0.001) significantly contributed to the total cost of the disease. The detail can be seen in Table 6.

The region-wise distribution of cost varies greatly, depending on factors, including accessibility of healthcare facilities and literacy rate and population’s financial condition. It was seen in the demographic distribution among four provinces of Pakistan, Punjab was having the highest cost of diabetes management. Among the Punjab region, the highest cost for diabetes management was observed in Federal Capital Territory with an average cost of USD 801. Second in line was Khyber Pakhtunkhwa with an estimated average cost of diabetes management at USD 765 with the highest cost reported in Peshawar city (average per person cost USD 797). The province of Sindh reported an average cost of USD 724, with Karachi region’s average cost at USD 773. In the province of Sindh, the estimated out of pocket cost for diabetes management was reported to be the lowest in the region, with an average cost of diabetes management at USD 675. Baluchistan province have the lowest average cost of diabetes management among all the provinces with USD 707 and Quetta city having an average cost of USD 723, Figure 2.

## 4. Discussion

COI analyses may provide helpful information for decision makers for distributing limited resources or desired amenities. Previous studies conducted in Pakistan were region-specific, and the cost component did not collect hospitalization data to evaluate the economic impact of diabetes care. These studies were limited to Karachi and South Punjab and did not provide enough information about the entire country. This study aimed to evaluate the nationwide economic consequences of diabetes. All attempts have been made in this study to assess all plausible cost components. Policymakers are interested in maintaining or lowering COI; hence, this study was primarily done to determine the cost of diabetes care in Pakistan. Individuals, families, and governments must allocate resources for health care based on existing evidence while also addressing other basic needs.

The current study was focused on the direct and indirect costs of diabetes management. In the present study, most patients were on oral anti-hyperglycemic drugs. The annual total cost of diabetes care was USD 740.1, amongst which the share of the direct cost was USD 646.4, and the indirect cost was USD93.65. The medicine (USD 274.5) and hospitalization (USD 319.7) shared the most direct cost. At the same time, the productivity loss of the patients had the highest contribution to the indirect cost (USD 81.36).

The calculation of diabetes-related costs has gained importance as the prevalence of this disorder increases with every passing day. In Pakistan, where healthcare resources are limited, it is imperative to calculate diabetes-related costs accurately. This will aid in the efficient allocation of healthcare resources. Direct costs are frequently higher than indirect costs, according to a thorough assessment published in 2015. Annual direct expenses ranged from USD 242 for an out-of-pocket study in Mexico to USD 11,917 for a diabetic cost study in the United States.

In contrast, indirect costs ranged from USD 45 in Pakistan to USD 16,914 in the Bahamas. In contrast to higher-income countries (HICs), patients in lower-middle-income countries (LMICs) bear a considerable financial burden through out-of-pocket medical payments. In a 2018 systematic analysis, the annual average cost of disease for T2D in low- and middle-income countries was estimated to be between USD 29.91 to USD 237.38. This difference may be affected by the different methods used in various studies [28]. The study reported from Bangladesh estimated the average annual cost of diabetes management was USD 864.7 per patient, which was in line with our study findings [29].

The other key finding of this study was that the patients hospitalized during the past year due to any complication related to diabetes incurred three times higher costs than those without hospitalization. The direct cost for the patient hospitalized due to diabetes-related complications was USD 1038. At the same time, medication and hospitalization contributed the most to the direct cost for those patients, with a mean cost of USD 432 and USD 419.4, respectively.

With a current diabetes prevalence of 16.98% and a total population of 220.9 million in Pakistan in 2020, the predicted direct cost of diabetes is 495.0 billion PKR, accounting for roughly 73.7 percent of Pakistan’s total federal and provincial yearly health expenditure (671.4 billion PKR). According to our research, diabetes treatment accounted for 23% of household earnings in the lowest-income groups. These findings are comparable to an Indian study, in which low-income diabetic individuals spent 25% of their monthly family income on diabetes care. Diabetes care is a critical impact on patients’ stress and the financial burden on the patients’ families and the government. Regarding socio-demographic characteristics, older patients, rural inhabitants, and those with a high socioeconomic status SES showed a significant link with direct diabetic expenditures. A prior study in Pakistan found a non-significant difference between age and direct medical costs [4]. Previous research from India and Singapore contradicted our findings; those investigations revealed no statistically significant differences between age and direct cost [30,31].

Regarding clinical parameters, direct costs increased dramatically as the disease duration and co-morbidities increased. Patients with one and more than one co-morbidity had 34.2 percent and 39.7 percent higher mean direct expenses, respectively, than patients without co-morbidities (*p* < 0.01). These findings are comparable to a cocoa study, which observed that a 91.8 percent increase in costs for patients with one or more co-morbidities (*p* ≤ 0.002) [32]. In both emerging and developed countries, similar tendencies have been observed.

In the current study, it was found that direct cost was the most common cost in the total cost of diabetes care. The share of the medicine was highly accounted for in the direct cost. The study reported from Kenya showed similar findings, where the treatment accounted for 52.4% of the total direct cost of diabetes care [33]. Similar results were reported from Bangladesh, where medicine contributed 60.7% of the total cost [29]. Moreover, in the current study, it was observed that the productivity loss of the patient was mostly accounted for by indirect cost. However, it was observed in the published literature that absenteeism, presenteeism, and premature mortality significantly contributed to indirect cost [34,35,36]. In addition, it was also noted that food and travel significantly contributed to the indirect cost [35].

A significant correlation was found between the socio-demographic parameter and their association with the overall cost of diabetes. The higher cost was reported in males, <60 years of the study population, with the highest cost of USD 1376.9. The study reported a higher cost associated with prolonged diabetes duration and higher HbA1c. Previous literature, considerably in low and middle-income countries, reported similar findings. It was reported that the higher cost of diabetes management was associated with the higher prevalence of comorbid conditions [37,38,39].

According to world bank data, the per capita GDP of Pakistan was USD 1193.73, as reported in 2020. The current study reported that the overall diabetes management cost was 62% of the per capita GDP of Pakistan. At the same time, Pakistan’s GDP per capita healthcare spending is at 3.38%. However, with such scarce spending on healthcare and the disease burden of diabetes in the region, it is high time for policymakers to consider universal health insurance coverage to minimize the overall diabetes management burden on the population [40,41].

Considering the overall cost of diabetes management to Pakistan, the population-based cost of illness computed, based on the prevalence of type 2 diabetes mellitus reported by the International Diabetes Federation of 33 million people [9], the total burden of diabetes management is USD 24.42 billion. The total burden of diabetes management is almost 1.60% of Pakistan’s GDP, as estimated from World Bank data for 2021. In the USA, the overall cost of diabetes management, estimated in 2017, was USD 327 billion. That is estimated to be 1.67% of the country’s total GDP [42]. It was documented that medicine prices is the significant contributor to the overall diabetes management cost. In lower-middle income countries including Pakistan there are challenges related to forming medicine pricing policies which ultimately led to higher medication cost. Accessibility to medicine has been essential in enhancing patient outcomes and lowering mortality on a worldwide scale. It is crucial to remember that drug costs play a significant role in increasing access to medications. Studies and empirical evidence have demonstrated that pricing for the same brand of medications vary greatly between nations [43].

## 5. Conclusions

The current study concluded that direct cost was one of the main contributing parameters to the total cost of diabetes management. The cost of medicines and hospitalization significantly contributed to the direct cost. In addition, patients’ loss of productivity was a remarkable contributor to the indirect cost. This study has significant implications for the government and policymakers to formulate a comprehensive healthcare plan or introduce a universal health coverage plan. The increasing diabetes prevalence is causing a substantial financial burden on the healthcare system, whether public or private. There is a need for a comprehensive diabetes care action plan, including the role of diabetes educators in the healthcare system, to spread awareness among the community.

## 6. Study Limitations

The study has some limitations in the estimation of direct and indirect costs. The cost calculation was done using monthly reported data and extrapolated to 12 months. Therefore, any monthly expenditure reported could be under- or overestimated for annual cost estimation. In addition to the direct and indirect costs, there was imperceptible cost linked to the deteriorated health-related quality of life. There was insufficient information to apply the fractional costing approach; therefore, the human capital approach was used to estimate the indirect cost of diabetes management. Currently, there is a lack of specific epidemiological data, which limits the estimation of attributable risk of morbidity, mortality and resource utilization. Thus, there is a dire need for new methodological approaches, which could evaluate the effect of comorbid conditions during cost estimation.

## Figures and Tables

**Figure 1 ijerph-19-12611-f001:**
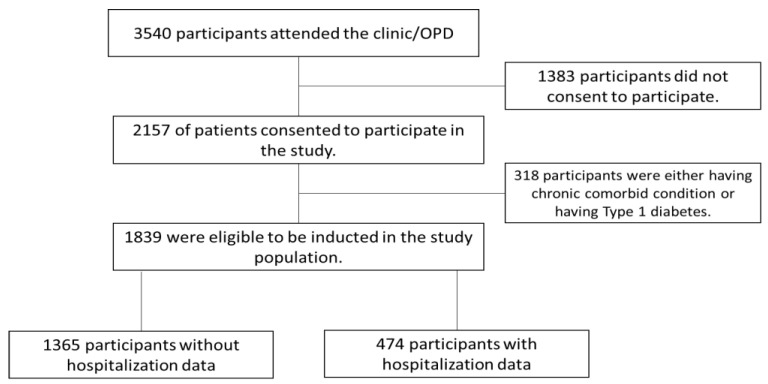
Schematic chart of diabetes participants recruitment in the study.

**Figure 2 ijerph-19-12611-f002:**
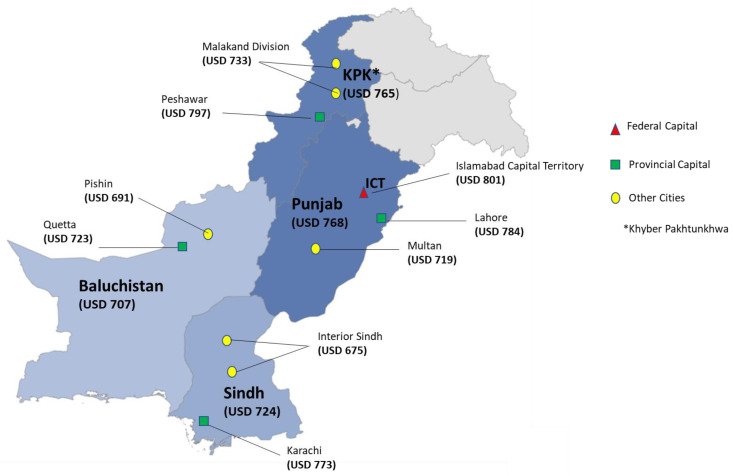
The cost comparison of the region computed per person diabetes management cost in USD.

**Table 1 ijerph-19-12611-t001:** Distribution of socio-demographic and clinical characteristics.

Variable	Number	Percentage
Age		
18–30	39	2.1
30–40	446	24.2
40–50	1109	60.1
50 and above	244	13.2
Gender		
Female	963	52.2
Male	876	47.5
Education status		
Illiterate	121	6.6
Graduated college and above	574	31.1
Intermediate (12 years of education or high school)	530	28.7
Primary	335	18.2
Secondary	271	14.7
Employment status		
Employed	859	46.6
Unemployed	986	53.4
Monthly Household Income		
≤100 USD	99	5.4
100–200	665	36
200–400	710	38.5
400–750	166	9
≤750	199	9.9
Marital Status		
Divorced	25	1.4
Married	1637	88.7
Separated	3	0.2
Single	174	9.4
Duration of Diabetes		
<1–5	1279	69.3
6–10	434	23.6
11–15	100	5.5
16 and above	27	1.4
Preferred Treatment		
Diet Plan/Exercise	37	2.01
Oral Hypoglycemic Agents	695	37.7
Combination of Oral and Insulin	823	44.8
Insulin Only	221	12.0
GLP-1 and NGI	63	3.4

**Table 2 ijerph-19-12611-t002:** Patients with diabetes in Pakistan: direct and indirect costs in USD.

Cost Components	Mean COI (USD)	Total COI * (Billion, USD)	% of Total COI
a. Direct medical cost			
Out-patient visit	16.5	0.544	2.2%
Hospitalization **	319.7	10.539	43.2%
Medicine	274.5	9.049	37.1%
Laboratory testing	12.3	0.405	1.7%
Miscellaneous Services ***	16.5	0.544	2.2%
Transportation	4.3	0.142	0.6%
Food	2.65	0.087	0.4%
Total direct cost	646.4	21.308	87.3%
b. Indirect cost			
Productivity loss of patient	81.36	2.682	11.0%
Caregiver Productivity loss	12.29	0.405	1.7%
Total indirect cost	93.65	3.087	12.7%
Total cost (a + b)	740.1	24.397	100.0%

* Total diabetes population cost of illness in billion USD computed for 32.9 million diabetics, according to IDF 2022. ** Those patients who got admitted to hospital for any relevant diabetes complication. *** Miscellaneous services include blood glucose testing, diabetes education service, etc.

**Table 3 ijerph-19-12611-t003:** Correlation among cost of diabetes management and sociodemographic varia.

Variables	Direct Cost (USD)		Indirect Cost (USD)	
	r	*p*-Value	r	*p*-Value
Gender	0.78	0.03	0.52	0.03
Duration of Diabetes	0.098	0.003	0.78	0.005
Education	−0.06	0.071	0.11	0.241
Monthly household income (USD)	−0.92	0.002	0.69	0.219

**Table 4 ijerph-19-12611-t004:** The cost of diabetes management and socio-demographic and clinical characteristics.

Variables	Direct Cost (USD)	*p*-Value	Indirect Cost (USD)	*p*-Value	Total Cost (USD)
	Mean		Mean		Mean
Gender *					
Male	545.2	<0.05 *	81.6	<0.05 *	761.9
Female	465.7		84.4		727.1
Age (years) **					
≤30	519.6	0.63	68.8	0.54	588.4
31–50	738.7		103.0		841.8
50–60	934.9		56.1		991.0
>60	1376.9		57.1		1434.1
Education **					
Illiterate	575.3	<0.05 *	61.6	0.08	636.9
Graduated college and above	659.5		59.3		718.9
Intermediate	874.1		88.0		962.1
Secondary	711.4		92.3		803.7
Primary	733.9		104.5		838.4
Work status *					
Unemployed	627.5	0.152	48.0	0.152	675.6
Employed	598.9		116.0		715.0
Area of residence **					
Rural	359.8	<0.05 *	61.7	0.517	421.6
Semi-urban	388.3		48.1		436.5
Urban	931.3		93.1		1024.4
Monthly household income (US-$) *					
≤250	539.6	<0.05 *	65.2	<0.05 *	604.9
251–750	692.1		64.6		756.8
751 and above	986.8		**138.0**		1124.9
Duration of Diabetes (in years) *					
≤5	526.3	<0.05 *	53.7	<0.05 *	580.1
6–10	645.9		61.4		707.4
≥11	1044.1		**116.7**		1160.8
Mode of treatment **					
Oral Antidiabetic Diabetics	476.2	0.128	49.9	0.261	526.17
Insulin Only	702.6		67.2		769.89
Insulin/GLP-1 + OAD	970.9		104.2		1075.20
Family history of diabetes *					
Yes	818.4	0.721	97.1	0.61	915.6
No	762.4		75.4		837.9
HbA1c (%)					
Good (≤6.9)	527.1	<0.05 *	84.5	0.189	611.7
Fair (7–7.9)	552.1		53.2		605.4
Poor (≥8)	567.9		58.5		626.5
Number of complications *					
None	466.8	<0.001 *	56.3	<0.05 *	523.2
**One or two**	657.5		81.9		739.4
**Three or more**	**1033**		117.1		1351.5
History of co-morbidity **					
**None**	595.5	0.588	63.8	0.895	659.3
**Cardiovascular**	870.1		94.3		964.4
**Dyslipidemia**	566.8		56.9		623.7
**HTN + Dyslipidemia**	**926.9**		**95.7**		1022.6

* Mann–Whitney U test ** Kruskal–Wallis test were done for group comparison; *p*-value was considered significant at *p* < 0.05.

**Table 5 ijerph-19-12611-t005:** The difference in the mean direct and indirect costs of diabetes care among hospitalized and non-hospitalized population.

Cost Components	Mean Cost (USD)	% of Total
Patient without hospital admission (*n* = 1365)		
a. Direct medical cost		
Out-patients call on	18.5	5.8%
Medicine	213.6	66.8%
Laboratory testing	23.5	7.4%
Miscellaneous Facilities	8.8	2.8%
Transportation	9.4	2.9%
Food	1.3	0.4%
Total direct cost	275.1	**86.1%**
b. Indirect cost		
Patient Productivity loss	27.4	8.6%
Caregiver Productivity loss	17.1	5.4%
Total indirect cost	44.5	**13.9%**
Total cost (a + b)	319.6	**100.0%**
Patient with hospital admission (*n* = 474)		
a. Direct medical cost		
Out-patients call on	12.3	1.2%
Hospitalization	419.4	40.4%
Medicine	432	41.6%
Laboratory testing	28.4	2.7%
Miscellaneous Facilities	12	1.2%
Transportation	15.5	1.5%
Food	4.7	0.5%
Total direct cost	924.3	89.0%
b. Indirect cost		
Patient Productivity loss	79.8	7.7%
Caregiver Productivity loss	34.6	3.3%
Total indirect cost	114.4	11.0%
Total cost (a + b)	1038.7	100.0%

**Table 6 ijerph-19-12611-t006:** Total cost analysis with the median regression.

Variables	Unadjusted		Adjusted	
	Coefficients	*p*-Value	Coefficients	*p*-Value
Gender with reference to Male				
Female	19.03	0.492	44.85	0.036
Age with reference to ≤30 years				
31–40 years	82.85	0.107	21.13	0.419
41–60 years	221.91	<0.001	2.86	0.930
≥60 years	741.58	<0.001	170.76	0.708
Mode of treatment with reference to OHA				
Insulin	140.69	0.042	65.40	0.260
Insulin + OHA	307.38	<0.001	152.87	<0.001
Duration of Diabetes with reference to ≤5 years)				
6–10	78.36	0.080	17.59	0.403
≥11	368.68	<0.001	66.93	0.025
HbA1c with reference to ≤ 6.9				
Fair (7–7.9)	45.42	0.216	−1.20	0.949
Poor (≥8)	79.41	0.009	22.50	0.406
History of co-morbidity				
Cardiovascular	151.55	<0.002	39.19	0.187
Dyslipidemia	−49.17	0.494	2.21	0.877
Hypertension + Dyslipidemia	170.1	0.001	49.2	0.101
Number of complications				
One or two	210.54	<0.001	63.69	0.003
Three or more	847.72	<0.001	440.93	<0.001

## Data Availability

The datasets used and analyzed during the current study are available from the corresponding author on reasonable request.

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
