# Peer review of "Cost of Illness Analysis of Type 2 Diabetes Mellitus: The Findings from a Lower-Middle Income Country"

_ijerph, 2022, doi:10.3390/ijerph191912611_

Round 1
Reviewer 1 Report
In this article the authors corelate the relation between T2D and the associated direct and indirect costs. While admittedly, this article addresses a global issue with the cost of the most common chronic disease of the world, the article is more suited to be published in a country-specific diabetic journal, in this case Pakistan. The authors have performed the analysis quite thoroughly to substantiate their claims. I have the following comments about the article that could be considered for a revision.
1) The study design needs to be represented in the manuscript in a schematic form and the screening criteria clearly laid out.
2) The questionnaire for the patients/clinicians must be attached in supplement to the article. It is completely unclear if the questionnaire reflect the outcomes represented by the parameters in this article.
3) Is it possible to associate the total cost per capita associated with T2DM management in the discussion? In perspective to the economy of Pakistan, I believe this will definitely strengthen the findings of this paper. This is essential as world wide inflation effect on rising cost of health care must be established.
4) The few under noted key findings of the article related to the socio economic demography associated with the costs and also number of complications associated need to be highlighted.
5)Finally, is it possible to present a comparison graph associating cities and rural areas from the data vs cumulative single value for the cost of diabetes? This could shed light into the effect of geographically most expensive areas. As the article is directed towards policy makers, I believe this can be very impactful.
Author Response
1) The study design needs to be represented in the manuscript in a schematic form and the screening criteria clearly laid out.
Response:
The schematic form and the screening criteria have been laid out as suggested in the revised manuscript.
2) The questionnaire for the patients/clinicians must be attached in supplement to the article. It is completely unclear if the questionnaire reflect the outcomes represented by the parameters in this article.
Response:
Thank you for the suggestion. The primary questionnaire has been added as a supplement to the revised manuscript.
3) Is it possible to associate the total cost per capita associated with T2DM management in the discussion? In perspective to the economy of Pakistan, I believe this will definitely strengthen the findings of this paper. This is essential as world wide inflation effect on rising cost of health care must be established.
Response:
Thank you for the comment. The association of the total cost with T2DM management have been discussed in the discussion section as suggested (lines 322-329).
4) The few under noted key findings of the article related to the socio economic demography associated with the costs and also number of complications associated need to be highlighted.
Response:
Thanks for the suggestion. The noteworthy findings of significant sociodemographic parameters associated with the costs have been added in the discussion section (lines 315-321).
5)Finally, is it possible to present a comparison graph associating cities and rural areas from the data vs cumulative single value for the cost of diabetes? This could shed light into the effect of geographically most expensive areas. As the article is directed towards policy makers, I believe this can be very impactful.
Response:
The description regarding the geographical association of different areas and their cost have been added in the revised manuscript. The graph also highlighted the average costs spread of the regions. (lines256-268)
Reviewer 2 Report
1) Title indicated this was a population-based study, but as samples were drawn from clinics and not the community, I find this inaccurate.
2) Avoid abbreviations in abstract (line 19)
3) Throughout the document, diabetes should not be capitalized.
4) The formats of in-text citations, tables and references are incorrect. Please check the journal's guidelines.
5) Introduction section is too brief and lacks flow. The section has to be rewritten with new evidence and elaboration.
6) line 76 - Vague description of sample size justification. Please elaborate use of power, alpha, software or formula.
7) section 2.2 - sampling can be aided with a flow chart
8) as the questionnaire was developed by the researchers, more details are needed (line 97-98)
9) section 3.2 - authors provided correlation values for categorical variables. this is incorrect.
10) Table 5 - Please elaborate the regression analysis. Earlier authors only mentioned regression analysis, but have presented median regression i.e quantile regression in spss. Why is this the case?
11) Study limitation?
Author Response
1) Title indicated this was a population-based study, but as samples were drawn from clinics and not the community, I find this inaccurate.
Response:
Thanks for the valuable comment. Sharing here that the sampling method includes the prevalence-based ratio of the population. Therefore, the title suggested the population-based survey. However, we welcome the reviewer to suggest a more appropriate title for the article.
2) Avoid abbreviations in abstract (line 19)
Response:
Thanks for highlighting this. The abbreviation has been removed from the abstract section.
3) Throughout the document, diabetes should not be capitalized.
Response:
Thanks for highlighting the mistake and the amendments have been made throughout the manuscript.
4) The formats of in-text citations, tables and references are incorrect. Please check the journal's guidelines.
Response:
Thank you for the valuable feedback. The formatting of in-text citations, tables and referencing have been corrected.
5) Introduction section is too brief and lacks flow. The section has to be rewritten with new evidence and elaboration.
Response:
Thank you for your comment. The introduction section has been revised with new pieces of evidence. Additionally, the region-specific details have been added in the introduction section too.
6) line 76 - Vague description of sample size justification. Please elaborate use of power, alpha, software or formula.
Response:
Thank you for the comment. The suggested information has been added in the section 2.2. Study population and sampling.
7) section 2.2 - sampling can be aided with a flow chart
Response:
Thank you for your comment. The schematic flow of sampling has been added in the method section as suggested in the revised manuscript..
8) as the questionnaire was developed by the researchers, more details are needed (line 97-98)
Response:
More details have been added in the revised manuscript (lines 137-147). In addition, the primary questionnaire has been added as a supplement material to the manuscript.
9) section 3.2 - authors provided correlation values for categorical variables. this is incorrect.
Response:
Thank you for your comment. In fact, the correlation shared in the section is the categorical variables and their relevant direct and indirect cost. The description of the section has explained the correlations of these categorical variables and the costing.
10) Table 5 - Please elaborate the regression analysis. Earlier authors only mentioned regression analysis, but have presented median regression i.e quantile regression in spss. Why is this the case?
Response:
Thanks for highlighting the mistake. It was a typographical error. There was a median regression statement missing in the statistical section. Amendment has been made in the revised manuscript.
11) Study limitation?
Response:
Thanks for highlighting it. The limitations section has been added to the revised manuscript.
Round 2
Reviewer 1 Report
Thanks to authors for incorporating all the reviewers suggestions. I recommend publishing the article in current form.
Author Response
Thank you for the comments and suggestions.
Reviewer 2 Report
1) The onus is on the authors to provide an accurate title, not on the peer reviewer. Population-based survey means you should have sampled for participants in community, not clinical setting. This is rather basic. Please amend your title to avoid confusion.
2) Section 3.2 - Correlations values are provided in text, but not in the table 3. Hence, I'm not able to evaluate if it was done correctly. No mention of correlation analysis in the Methods (section 2.5.3.) as well.
3) You have to provide some details on the median regression as to assumption taken and adjustments made in the Methods (section 2.5.3)
Author Response
1) The onus is on the authors to provide an accurate title, not on the peer reviewer. Population-based survey means you should have sampled for participants in community, not clinical setting. This is rather basic. Please amend your title to avoid confusion. \
Response:
Thank you for the comment. The title has been modification with the new title as follows. “Cost of Illness Analysis of Type 2 Diabetes Mellitus: The findings from a Lower-Middle Income Country”.
2) Section 3.2 - Correlations values are provided in text, but not in the table 3. Hence, I'm not able to evaluate if it was done correctly. No mention of correlation analysis in the Methods (section 2.5.3.) as well.
Response:
The details for the correlation values have been provided as suggested in table 3. The details regarding the correlation analysis has been provided in the revised manuscript (Lines 196-198).
3) You have to provide some details on the median regression as to assumption taken and adjustments made in the Methods (section 2.5.3)
Response:
Details regarding median regression have been provided in the methodology section as suggested (Lines 198-202).
